# Remodeling of Perineuronal Nets in the Striato-Cortical Axis in L-DOPA-Induced Dyskinesia Rat Model

**DOI:** 10.3390/ijms262311726

**Published:** 2025-12-03

**Authors:** Nedime Tugce Bilbay, Banu Cahide Tel, Gulsum Akkus, Canan Cakir-Aktas, Taha Solakoglu, Gul Yalcin-Cakmakli, Bulent Elibol

**Affiliations:** 1Institute of Neurological Sciences and Psychiatry, Hacettepe University, 06100 Ankara, Türkiye; c.cakiraktas@hacettepe.edu.tr (C.C.-A.); tahasolakoglu@g.ucla.edu (T.S.); bulentelibol@gmail.com (B.E.); 2Neurology Department, Ministry of Health Diskapi Yildirim Beyazit Training and Research Hospital, 06110 Ankara, Türkiye; 3Department of Pharmacology, Faculty of Pharmacy, Hacettepe University, 06100 Ankara, Türkiye; banutel@hacettepe.edu.tr (B.C.T.); gulsummakkus@gmail.com (G.A.); 4Department of Neurology, Faculty of Medicine, Hacettepe University, 06100 Ankara, Türkiye; gulyalcin@yahoo.com

**Keywords:** Parkinson’s disease, L-DOPA induced dyskinesia, perineuronal nets, parvalbumin interneuron, WFA, ChABC, 6-OHDA rat model, striatum, cortex

## Abstract

L-DOPA-induced dyskinesia (LID) remains the most challenging complication of dopamine replacement therapy in Parkinson’s disease, correlated with maladaptive plasticity within corticostriatal circuits. Perineuronal nets (PNNs), extracellular matrix structures enwrapping mainly parvalbumin interneurons (PV-INs), are key regulators of neuronal stability and plasticity, yet their contribution to LID is unknown. Using a unilateral 6-hydroxydopamine rat model of Parkinsonism followed by chronic L-DOPA administration, we quantified PNN–PV associations by Wisteria floribunda agglutinin (WFA) and PV immunolabeling across striatal and motor cortical territories. Dopamine loss markedly reduced PNN density and intensity in the dorsolateral striatum (DLS), which only partially recovered after L-DOPA. In LID, canonical WFA^+^/PV^+^ cells remained low, whereas non-canonical WFA^−^/PV^+^ populations expanded in both DLS and M1 motor cortex (M1), indicating region-specific remodeling toward a high-plasticity state. To assess causality, we used Chondroitinase ABC (ChABC) for PNN degradation. DLS-targeted ChABC exacerbated abnormal involuntary movements and increased local PV density, while M1-ChABC had no behavioral effect but altered PV metrics within the DLS–M1 axis. These findings identify the DLS as a critical node where PNN fragility amplifies dyskinesia, highlight a functional coupling between striatal and cortical PNN–PV remodeling, and suggest that stabilizing extracellular matrix integrity could mitigate maladaptive plasticity underlying LID.

## 1. Introduction

Parkinson’s disease (PD) is the second most prevalent neurodegenerative disorder worldwide, surpassed by Alzheimer’s disease [1,2]. For over six decades, the dopamine precursor L-3,4-dihydroxyphenylalanine (L-DOPA) has remained the gold-standard medication for alleviating the motor symptoms of PD. Yet, with chronic administration, L-DOPA frequently gives rise to various forms of involuntary movements collectively termed L-DOPA-induced dyskinesias (LIDs) [3].

It is well accepted that LIDs are caused by the combination of nigral dopaminergic cell loss and chronic pulsatile dopaminergic receptor stimulation [4]. The emergence of LID is increasingly understood as the consequence of maladaptive plasticity within the neuronal circuit between the motor cortex and the striatum [5]. At the synaptic level, converging evidence points to structural and molecular perturbations across presynaptic, postsynaptic, and glial compartments—the core elements of synaptic transmission [6,7]. It is known that the extracellular matrix (ECM) is also involved in synaptic homeostasis, remodeling, and plasticity [8]. The ECM contains three main structural components: the basement membrane, the perineuronal nets (PNNs), and the interstitial matrix, which are collectively considered the fourth component of the ‘tetrapartite synapse’ organization [9].

Among the extracellular matrix structures, PNNs are of particular interest. PNNs consist mainly of chondroitin sulfate proteoglycans (CSPGs) held together by link proteins to a hyaluronan backbone, forming a rigid, honeycomb-like lattice around distinct classes of neurons throughout the brain and spinal cord. PNNs appear during critical periods of development by promoting synaptic stabilization and limiting brain plasticity in adulthood [10,11,12,13,14,15]. PNNs predominantly ensheathe the soma and proximal dendrites of fast-spiking parvalbumin-expressing interneurons (PV-INs) [16,17], PV-INs, which form a highly interconnected network in the cortex, inhibit nearby pyramidal cells, and generate gamma frequency rhythmic oscillations, which are important for information processing [18,19,20]. By stabilizing PV-IN excitability, PNNs exert a crucial homeostatic influence on microcircuit dynamics, namely the excitation–inhibition (E–I) balance, and have been typically implicated in cortical sensory plasticity and memory modulation [21,22,23,24,25,26]. PV-INs are also the predominant subtype of gamma-aminobutyric acidergic INs in the striatum; they receive strong cortical and thalamic excitatory inputs and control the activity patterns and plasticity of striatal spiny projection neurons via powerful feedforward inhibition [27,28,29].

PNNs, by regulating and hampering new excitatory and inhibitory connections to PV-INs, make learning and acquiring new skills difficult during adult life, but they are dynamic in nature and open to various modifications, which may have important consequences in health and disease [14,30,31,32]. Indeed, the transient removal of PNNs in the cortex by local injections of the chondroitinase ABC (ChABC), a bacterial lyase which degrades CSPG side chains, reverses PV-IN maturation and enhances adult plasticity [25,33,34]. ChABC has been widely used in the adult primary visual cortex to restore binocular vision in rodent models of amblyopia, and also in models of spinal cord injury with positive behavioral outcomes [35,36]. Disruption of PNN integrity has also been implicated in a wide spectrum of neurological and psychiatric conditions, including disorders associated with drug abuse [37,38,39], fear [40,41], depression [42], Alzheimer’s disease [43,44], epilepsy [45], stroke [46], autism [47,48], and schizophrenia [49,50].

However, the studies exploring the role of PNN-related PV-IN activity change in the acquisition of physiological and disordered motor behavior, particularly in models of Parkinsonism and LID, are very limited. Although there is compelling evidence indicating the possibility of remodeling PV-IN-centered microcircuits both in motor striatal and cortical regions, which in turn contributes to well-recognized network-level pathological oscillations during dopaminergic deprivation and replacement therapies resulting in LIDs [51,52,53,54], the plausible involvement of PNNs in these dynamics still remains elusive. In a recent study, it was found that PNN levels were transiently decreased in the primary motor cortex in a unilateral 6-hydroxydopamine (6-OHDA) model of PD. Furthermore, PNN degradation by bilateral ChABC injections into M1 cortices led to partial motor recovery when coupled with motor stimulation, suggesting motor cortical PNN modulations might have an implication in PD models [55]. In the only study addressing the role of striatal PNN integrity in abnormal motor behaviors, excessive repetitive behavior was found to be related to increased density of PNN-positive PV-INs in dorsomedial striatum (DMS) in several mouse models tested, and degradation of PNN by ChABC normalized this behavior [56].

Based on these premises, we hypothesized that alterations in PNNs, and consequently, the activity of PV-INs, may contribute to the maladaptive synaptic changes underlying LID. To test this hypothesis, we employed the unilateral 6-OHDA rat model of Parkinsonism and subsequently induced dyskinesia through chronic L-DOPA administration. We then comparatively performed cytometric analysis for PNN-enwrapped and PV-expressing neurons in dorsolateral, dorsomedial, and ventral territories of striatum together with M1 and M2 sectors of motor cortex, both in Parkinsonism and dyskinetic conditions, measured by motor tests and detailed abnormal involuntary movements (AIMs) scoring. We have found significant reductions in PNN intensity and PNN-positive PV-IN density after dopaminergic deprivation, mainly in the dorsolateral striatum (DLS) and M1 motor cortex (M1), which were partially restored after L-DOPA treatment. However, there was a consistent increase in PNN negative PV-IN density in both regions, suggesting a PNN remodeling in subsets of PV-INs during LID development. Causality assessment by using targeted ChABC injections into the DLS and M1 revealed that PNN degradation in DLS exacerbated AIMs scores, whereas M1 injections had no significant effect.

## 2. Results

### 2.1. Open-Field and Cylinder Tests Revealed a Behavioral Profile Consistent with 6-OHDA-Induced Motor Deficits

On post-injection week 3, striatal denervation was assessed using the open field, cylinder, and apomorphine-induced rotation assays (Figure 1A). In the open-field test, total distance traveled was reduced in the 6-OHDA group compared with sham and naïve controls, whereas stereotypy, ambulatory activity, rest percentage, and horizontal activity counts remained unchanged (Appendix A). In the cylinder test, left forepaw use was markedly reduced in the 6-OHDA group relative to both controls (Appendix A). Together, the decreased locomotor output and forelimb asymmetry confirmed unilateral dopaminergic denervation in this model.

### 2.2. AIMs Increased over Time, with Limited Early Changes and a Marked Rise at Later Time Points

After behavioral baseline testing, rats received daily L-DOPA/benserazide (25/6.25 mg/kg) for 15 days to induce LID (Figure 1A). For total AIMs, days 1 and 4 were comparable. By day 13, total AIMs were higher than on day 4 (Appendix A). Locomotor scores were also higher on day 13 than on day 1 and 4 (Appendix A). Axial-limb-orolingual (ALO) and limb subscores showed the same trajectory (Appendix A). Orolingual scores were unchanged across days (Appendix A), whereas axial scores differed between days 1 and 7 (Appendix A). On day 13, we observed a modest peak near 40 min, a plateau to about 100 min, and a subsequent drop by 120 min (Appendix A).

### 2.3. In DLS, PNN–PV Profiles Shifted from Canonical to Non-Canonical States in the LID Group

After the behavioral assays, we assessed group differences in PNN–PV association within the DLS, the sensorimotor territory of the striatum (Figure 2A). Total Wisteria floribunda agglutinin (WFA) positive (WFA^+^) cell density was reduced in the Parkinsonism group relative to sham and naïve controls and exhibited partial recovery toward control levels in the LID group (Figure 2B, Appendix A). By contrast, WFA intensity remained lower in both Parkinsonism and LID groups compared to the naïve group (Figure 2G, Appendix A). Together, these data indicate that L-DOPA partially restores PNN-positive cell density, whereas PNN intensity remains below the levels of naïve animals, suggesting sustained capacity for plastic change within DLS microcircuits.

In the Parkinsonism group, total PV-positive (PV^+^) cell density tended to be lower compared to both control groups (Figure 2C, Appendix A). Following L-DOPA treatment, PV^+^ cell density increased significantly.

In line with density changes, PV intensity followed a similar trend (Figure 2H, Appendix A). In the Parkinsonism group, mean PV intensity showed a decreasing tendency relative to controls but did not reach significance. Contrary to expectations, L-DOPA treatment failed to reverse this reduction; instead, PV intensity was significantly lower in the LID group compared to naïve animals. Overall, PV intensity values in both pathological groups remained below control levels, with statistical significance only in the LID group (Figure 2H, Appendix A), suggesting that PV-INs are sensitive to dopaminergic depletion; notably, L-DOPA did not normalize this alteration in our model.

Given the increase in total PV^+^ cell density in the LID group, we next analyzed cellular subpopulations to determine whether the PNN–PV association underwent dynamic reconfiguration (Figure 2D–F, Appendix A). In the naïve group, ~93% of WFA-expressing cells were PV-IN positive (WFA^+^/PV^+^) (Figure 2D, Appendix A), forming the predominant canonical subpopulation. A small fraction (~7%) of WFA-labeled cells lacked PV expression (WFA^+^/PV^−^), which indicates a minor population whose precise neuronal identity within the striatum remains uncertain [57] (Figure 2E, Appendix A). On the other hand, about 69% of PV-INs were enwrapped by WFA labeling (WFA^+^/PV^+^), while the remaining ~31% lacked WFA (WFA^−^/PV^+^) and represented the non-canonical subpopulation (Figure 2F, Appendix A).

Canonical WFA^+^/PV^+^ cell density was significantly reduced in the Parkinsonism group relative to the sham and naïve groups and remained significantly lower in the LID group compared to the sham group (Figure 2D, Appendix A).

In contrast, the WFA^+^/PV^−^ subpopulation, although numerically minor, increased in the LID group relative to all other groups (Figure 2E, Appendix A), with unclear functional significance. However, the non-canonical WFA^−^/PV^+^ population was markedly elevated in the LID group compared to the other groups (Figure 2F, Appendix A), leading to a prominent shift from canonical to non-canonical remodeling, which might play a role in LID development.

### 2.4. All Cellular Metrics Remained Unchanged in DMS, Whereas Ventral Striatum (VS) Showed Limited Alterations

After characterizing the sensorimotor DLS, we evaluated the DMS and the VS, which serve associative and limbic functions, respectively (Appendix A, Appendix A). In DMS, no significant differences were detected across any PNN–PV association metrics (Appendix A, Appendix A). Similarly, in the VS, total WFA and PV cell densities, as well as canonical and non-canonical subpopulations, showed no significant changes (Appendix A, Appendix A). However, in parallel with our findings in the DLS, the density of WFA^+^/PV^−^ cells increased in the LID group relative to the Parkinsonism group (Appendix A, Appendix A), though the relevance of this change remains uncertain. Additionally, total PV intensity was lower in the Parkinsonism group compared to both naïve and sham controls (Appendix A, Appendix A). PV intensity was also reduced after L-DOPA treatment with a significant difference relative to the naïve group (Appendix A, Appendix A), whereas total WFA intensity was unchanged (Appendix A, Appendix A). Taken together, all these data indicate that PNN–PV dynamics are not homogeneous across the striatum, with remodeling most pronounced in the DLS.

### 2.5. In M1 Under LID Condition, Total PV-IN Cell Density Rose, Driven by an Increase in the Non-Canonical WFA^−^/PV^+^ Subpopulation

We then evaluated the PNN–PV association in the M1 to probe corticostriatal plasticity in LID (Figure 3A). Total WFA^+^ cell density and intensity did not differ among groups (Figure 3B,G, Appendix A). By contrast, total PV^+^ cell density was higher in the LID group compared to naïve animals (Figure 3C, Appendix A), while total PV intensity was unchanged (Figure 3H, Appendix A), indicating a remodeling of the PV^+^ cell population in the LID condition.

When cellular subpopulations were comparatively analyzed (Figure 3D–F, Appendix A), approximately 97% of WFA-labeled cells in the M1 of naïve animals were PV-expressing cells (Figure 3D, Appendix A), whereas the remaining ~3% lacked PV expression (Figure 3E, Appendix A). Among PV-expressing cells, ~91% were WFA positive (Figure 3D, Appendix A), while the non-canonical WFA^−^/PV^+^ subpopulation accounted for ~9% (Figure 3F, Appendix A). Canonical WFA^+^/PV^+^ cells and WFA^+^/PV^−^ subpopulations did not differ significantly across groups (Figure 3D,E, Appendix A). In contrast, the non-canonical WFA^−^/PV^+^ cells markedly increased in the LID group compared to the naïve and Parkinsonism groups (Figure 3F, Appendix A). Consistent with our DLS findings, the L-DOPA-associated increase in the WFA^−^/PV^+^ cell subpopulation mirrors the rise in total PV density in M1 and suggests a potential contribution of this particular cell group to LID development.

### 2.6. Secondary Motor Cortex (M2) Exhibited Restricted Changes Confined to PNN Intensity

Following the M1 analysis, we examined the M2, which has been implicated in the regulation of motor output via corticostriatal pathways [58,59]. While all other PNN–PV association metrics remained unchanged (Appendix A, Appendix A), total WFA intensity was higher in the sham and LID groups compared to the naïve group, whereas the Parkinsonism group showed a non-significant increase relative to the naïve group (Appendix A, Appendix A). The precise interpretation of these findings remains uncertain, but it may be related to surgical intervention rather than disease-specific effects.

A combined analysis of striatal and cortical findings revealed that PNN–PV remodeling in LID was topographically segregated, with changes most pronounced in DLS and M1. Accordingly, DLS and M1 were designated as target regions for ChABC to enzymatically degrade the PNN structures (DLS-ChABC and M1-ChABC). Then, we assessed behavioral effects and the cellular metrics both at the local injection site and at the corresponding projection region within the DLS–M1 axis (Figure 1B).

### 2.7. Total AIMs Was Exacerbated by DLS-ChABC Injection, but Were Not Altered by M1-ChABC Administration

Following ChABC administration into either DLS or M1, we started L-DOPA treatment on post-ChABC day 3, and dyskinesias were evaluated every other day for 13 days (Figure 1B). Total AIMs were consistently higher in the DLS-ChABC group across all scoring days, with significant elevations on days 3 and 5 (Figure 4A). Subscoring of the total AIMs revealed that the locomotor component did not show any change throughout the testing days (Figure 4B), while the ALO subscores were consistently higher on days 3, 5, 9, and 13 (Figure 4C). Notably, the elevation of ALO subscores appeared to arise mainly from the orolingual component, which showed a similar increase on days 5, 9, 11, and 13 (Figure 4D). In contrast, the limb component of ALO did not differ across all scoring days, while the axial component was elevated only on day 5 (Figure 4E,F). Together, these data indicate that DLS-targeted PNN degradation enhances dyskinesia primarily through orolingual manifestations.

On the contrary, M1-targeted ChABC injection did not lead to any change in either total AIMs or its component subscores between study groups (Figure 5A–F). Taken together, LIDs rose with ChABC application into the DLS but not into M1, highlighting the greater contribution of DLS as a dyskinesia generator.

### 2.8. DLS-ChABC Administration Increased PV Density in DLS and Reduced WFA and PV Intensity in M1

Following the behavioral readouts, we examined local PNN–PV association in DLS on post-injection day 16 (on day 13 of L-DOPA treatment) (Figure 1B). Total PV^+^ cell density was higher in the DLS-ChABC group compared to the vehicle group (Figure 6B, Appendix A), while all other parameters were unchanged (Figure 6A,C–G, Appendix A). In the cross-region M1, PNN–PV densities were unchanged (Figure 6H–L, Appendix A), whereas intensities of WFA (Figure 6M, Appendix A) and PV were lower in the DLS-ChABC-injected group (Figure 6N, Appendix A).

### 2.9. M1-ChABC Administration Increased PV Intensity in M1 and Expanded PV Parameters in DLS

In the M1-ChABC condition, we profiled PNN–PV association after the behavioral assessments. Despite no change in AIMs, some cellular alterations emerged both locally and in the cross-region DLS. Locally (M1), total PV intensity was higher in the M1-ChABC group compared to the vehicle group (Figure 7B, Appendix A), with no differences in other measures (Figure 7A,C–G, Appendix A).

Remarkably, in the cross-region (DLS) we observed an increase in total PV^+^ cell density (Figure 7I, Appendix A), accompanied by an expansion of the non-canonical WFA^−^/PV^+^ subpopulation in the ChABC-injected group (Figure 7L, Appendix A). Likewise, PV intensity increased (Figure 7N, Appendix A), whereas the remaining measures showed no differences (Figure 7H,J,K,M, Appendix A).

## 3. Discussion

LID, the most troublesome complication of dopamine replacement therapy, remains a major management challenge in clinical practice. Although substantial experimental efforts have been made across micro- and macro-circuit scales in the cortico-basal ganglia network, the underlying pathophysiological mechanisms are not fully depicted [6,60]. Here, we provide an initial, morphology-based assessment of the potential contribution of PNN-associated PV-IN remodeling in LID development in the striatum and motor cortex.

Our findings show that unilateral dopaminergic deprivation followed by chronic L-DOPA exposure drives region-specific remodeling of the PNN–PV-IN associations across different striatal territories. There was a significant reduction in WFA intensity in DLS, which remained persistently diminished during both Parkinsonism and LID conditions, while there was little or no change in DM and VS regions. In parallel, at the cellular level, the WFA^+^ cell density was also reduced in dopamine-depleted DLS, which was partly normalized after L-DOPA treatment. This pattern indicates that while PNN density partially recovers by dopaminergic replacement, PNN structural integrity fails to fully do so, leaving the DLS circuits in a pro-plastic state.

As was expected, the majority of WFA^+^ striatal cells were PV-INs, which are critical for all adaptive and maladaptive plastic changes in all regions of striatal microcircuits [57]. We have found PV^+^ cell density in DLS tends to be reduced in Parkinsonism, which was normalized and even slightly increased after L-DOPA treatment. This trajectory aligns with previous reports that described reduced expression and early synaptic dysfunction in striatal PV-INs under dopamine loss [52,61,62], and they were consistent with the findings of Gomez et al., showing that L-DOPA administration partially restored reduced PV-IN density in 6-ODHA-injected animals [62]. At the subpopulation level, WFA^+^/PV^+^ cells dropped to their lowest levels in Parkinsonism and tended to recover after L-DOPA treatment, suggesting that dopamine loss weakens the canonical PNN–PV-IN association and constrains recovery potential. By contrast, there was a significant increase in the density of WFA-negative PV-IN subpopulation in DLS, indicating the non-canonical pool expanded in the LID condition.

The phenotypic and physiological characteristics of these cells are currently unknown: cortical PV-INs comprise multiple subtypes with distinct structural, neurochemical, and physiological features [63,64], while striatal PV-INs, particularly enriched within the DLS, are usually known as a functionally homogenous population [57,65,66]. There is, however, some evidence for physiological and molecular heterogeneity, suggesting that distinct subpopulations of PV-INs also exist within the striatum [67]. Although still elusive, different patterns of PNN density may well be a part of this heterogeneity; indeed, there are a considerable number of striatal PV-INs, which are devoid of PNN [57]. The significant rise in WFA^−^/PV^+^ cell density in DLS may likely be related to the degradation of already poor PNN structures enwrapping a subset of PV-INs and point to a change into a functionally high-plasticity state in the LID condition. As elegantly reviewed, most in vivo and in vitro studies indicated that PNN loss or degradation reduced the firing rates of PV-INs in adult cortical networks and returned them to a juvenile-like state [14]. Taken together, these observations suggest that the malleable PNN scaffold (indexed by WFA positivity) tunes the PV-IN cell plasticity; therefore, a shift from canonical to non-canonical PNN–PV-IN configuration in DLS might fuel maladaptive remodeling in LID development.

On the other hand, in the DMS and VS, no significant changes were detected in these metrics, whereas in the VS, the WFA^+^/PV^−^ subpopulation was found to be elevated in the LID condition, as was detected in the DLS. Although constituting a very small proportion of total WFA^+^ cells in the striatum, and their phenotypic characteristics are not fully defined, previous data indicate that they are neither choline acetyltransferase^+^ nor calretinin^+^ INs and may correspond to somatostatin^+^ INs [57]. Given the fact that increased PNN expression points to more stable synaptic activity and plasticity, further work needs to be performed to explore their role in LID development. Taken together, these results indicate that the PNN–PV dynamics are not homogeneous throughout the striatum; the DLS—a region tightly linked to motor control and dyskinesias—emerges as the primary locus of remodeling, whereas DMS and VS exhibit limited changes.

To probe cortical correlates of the PNN–PV remodeling in striatum, we examined the same cellular metrics in both M1 and M2 motor regions throughout all their cortical layers. In M1, the primary motor cortex, total WFA^+^ cell density and WFA intensity were unchanged across all study groups. This cortical profile is compatible with a recent report of early loss but later normalization of WFA density (~5-week post-lesion) in the M1 motor cortex of the 6-OHDA-induced Parkinsonism model in mice [55]. We have found that PV cell density and intensity were also unchanged in Parkinsonism at all layers of M1. Likewise, it was recently shown that loss of midbrain dopaminergic cells did not alter the number, morphology, and physiology of PV-INs in layer 5 of M1 cortex in mice [68], which was found to be consistent with earlier findings in postmortem human M1 [69], although another study had previously described increased PV immunoreactivity in the same region [70]. Similarly to our findings in M1, we found no change in WFA^+^ and PV^+^ cell densities and subpopulation compositions in M2, which corresponds to the supplementary motor area in rodents [71]. These findings contrast with a recent report that observed decreased PV-IN density in layers II/III of both M1 and M2 cortices in a mouse model of Parkinsonism [59]. These discrepancies likely stem from differences in the cortical layers examined, model type, and timing of assessment.

On the other hand, we found that PV-IN cell density slightly increased without a change in PV intensity in M1 but not in M2 after L-DOPA treatment, interestingly, with a disproportionate rise in the non-canonical WFA^−^/PV^+^ cell density, similar to our findings in the DLS. In this framework, elevation of non-canonical WFA^−^/PV^+^ cell density in the DLS-M1 axis after L-DOPA treatment might be related to the generation of narrow-band high-γ oscillations, which were strongly suggested in LID models [72,73,74]. This exciting possibility opens a new avenue for future work, exploring the relationship between PNN remodeling and narrow-band high-γ activity in the process of LID development.

PNN fragility on PV-INs in the striato-cortical axis is plausibly driven by intrinsic and/or extrinsic factors determining the synthesis and degradation dynamics of PNN in both Parkinsonism and LID conditions. In the adult mouse visual cortex, chemogenetic inhibition of PV-INs at the single-cell level induces PNN regression. This demonstrates that the genetic makeup of PV-INs allows them to individually regulate their own PNN density in an activity-dependent manner [75]. Alternatively, neuroinflammation and microglial activation associated with neuronal degeneration may also provide a mechanistic basis for fragility and/or incomplete recovery of PNN [76]. Indeed, several groups showed a strong correlation between LID development and the activation of microglial subpopulations in the striatum [59,77,78,79,80]. In all these studies, it was suggested that microglial activation may have a role in LID pathophysiology by being involved in the renowned synaptic rearrangements of local microcircuits. However, considering their well-known catabolic role [81,82,83], microglial activation might well be critical in the pathological degradation of PNNs leading to LID development.

Based on these findings, we concentrated our causality tests on modulating the plasticity at the level of PNN by applying ChABC to DLS and M1, where PNN remodeling was found to be most prominent. This design enabled region-resolved comparisons between behavioral readouts (AIMs) and cellular metrics and allowed us to test the circuit-level contributions of PNN perturbation. Indeed, the ChABC application into the DLS produced clear behavioral effects, whereas M1-ChABC yielded no change. ChABC delivery to the DLS increased LID plasticity, elevating total AIMs on days 3 and 5 (corresponding to post-ChABC days 6 and 8). This profile supports day-to-week-scale persistence of PNN digestion effects, followed by gradual reconstitution [26,55,84,85]. Among AIMs components, locomotor scores were unchanged, whereas the ALO increase persisted until day 13, carried predominantly by orolingual and to some extent by axial subscores. At the circuit level, the sensorimotor DLS is central to LID pathophysiology, while locomotor measures can reflect broader striatal functions engaging DMS [56,86,87]. Taken together, these data suggest that ALO—the parameter closest to human dyskinesias—can be modulated by decomposing the integrity of striatal PNN structures, which needs to be explored by more specific molecular and electrophysiological techniques.

At the cellular level, however, following ChABC administrations, PNN–PV metrics showed composite changes across the DLS-M1 axis, partly matching the observed behavioral effects. Since the plausible compensatory changes in PNN synthesis and the nonspecific catalytic effects of ChABC would confound the cellular metrics in injected regions, the assessments in their corresponding output areas might give more reliable insights. By saying this, only the total PV density was found to be slightly increased in the DLS after ChABC injection. However, we have found that WFA and PV intensities decreased in M1, the cross-region of DLS, indicating a shift towards maturation-level weakening of cortical PNNs, consistent with the augmented LID development. Similarly, in M1-ChABC-injected animals, where no behavioral change was induced, all cellular metrics were stable, except for a slight increase in PV intensity in the targeted M1 region. Interestingly, however, we observed increased PV intensity with a significant rise in the non-canonical WFA^−^ PV-IN subpopulation in the cross-region DLS. This indicates that cortical PNN attenuation can propagate cellular signatures to the striatum, yet does not translate into behavior within our time window. When both ChABC applications were considered together, injection-site PNN metrics were not significantly changed, while PV parameters showed region-specific remodeling, implicating site- and time-dependent ChABC effects. Behaviorally, the augmentation of LID with DLS, but not with M1-ChABC application, suggests the greater contribution of DLS to the dyskinesia generation through the DLS-M1 axis.

These interpretations should be viewed in the light of our single sampling window (post-ChABC day 16), which is the main limitation of our study design. At this time point, injection-site PNN readouts (WFA^+^ cell density and intensity) were unchanged, which likely reflects reconstitution/recovery after an earlier degradation phase. A single-time point design cannot discriminate between genuine region-specific differences and phase offsets in PNN metabolism and reassembly across areas; multi-time point sampling will be required to resolve these divergences and their temporal alignment with behavior.

In conclusion, for the first time, our findings suggest that PNN–PV-IN axis may be an active contributor to maladaptive plasticity in LID, manifested by reduced WFA expression and a cell-state shift from canonical (WFA^+^/PV^+^) toward non-canonical (WFA^−^/PV^+^) populations. This morphological remodeling is regionally patterned across interconnected striatal and cortical territories, namely DLS and M1 motor cortex. Causality tests using ChABC-mediated PNN degradation further indicate that the DLS acts as the node that initiates and/or sustains the behavioral phenotype in LID, whereas M1 seems largely to contribute to macro-circuit-level pathological organization in LID. All together, these results argue that strategies aimed at stabilizing PNN integrity and rebalancing the PNN–PV-IN associations may hold translational promise for mitigating LID development.

## 4. Materials and Methods

### 4.1. Experimental Animals

Sixty-four male Wistar rats (230–270 g, 6–8 weeks) were obtained from the Kobay Experimental Animals Laboratory (Ankara, Türkiye). Rats were maintained under stable environmental conditions (18–22 °C, constant humidity, 12 h light/dark cycle) with ad libitum access to food and water. All procedures were conducted in accordance with institutional guidelines and approved by the Hacettepe University Animal Experiments Local Ethics Committee (2020/11-02 and 24 December 2020).

### 4.2. Pharmacological Agents

6-OHDA (12.5 µg in 5 µL of 0.02% ascorbic acid; Cayman Chemical, Ann Arbor, MI, USA), and ChABC (50 U/mL prepared in 0.1% bovine serum albumin (BSA) in phosphate-buffered saline (PBS); Sigma-Aldrich, St. Louis, MO, USA) were used for lesioning and PNN degradation, respectively. L-DOPA (25 mg/kg, intraperitoneal, i.p.; Cayman Chemical, Ann Arbor, MI, USA) and benserazide hydrochloride (6.25 mg/kg, i.p.; Cayman Chemical, Ann Arbor, MI, USA) were freshly dissolved in sterile distilled water immediately prior to injection. Apomorphine hydrochloride hemihydrate (0.5 mg/kg, i.p.; Sigma-Aldrich, St. Louis, MO, USA) and desipramine hydrochloride (25 mg/kg, i.p.; Sigma-Aldrich, St. Louis, MO, USA) were also used. For anesthesia/analgesia, ketamine (80 mg/kg, i.p.; Pfizer Inc., New York, NY, USA), xylazine (10 mg/kg, i.p.; Bioveta a.s., Ivanovice na Hané, Czech Republic), prilocaine (1–2 mg/kg, intradermal, i.d.; Vem İlaç, İstanbul, Türkiye), and flunixin (2.5 mg/kg, subcutaneous, s.c.; Vilsan İlaç, Ankara, Türkiye) were administered according to standard protocols.

### 4.3. Experimental Design and Timeline

Two complementary experimental paradigms were employed (Figure 1A). In the first design, animals were allocated to four cohorts: naïve (n = 8), sham-operated (n = 8), 6-OHDA-induced Parkinsonism (n = 8), and LID (n = 8). Nigrostriatal denervation was induced by stereotaxic injection of 6-OHDA into the right medial forebrain bundle (MFB), while sham animals received the vehicle (ascorbic acid). Lesion efficacy was confirmed on days 20 and 21 using a battery of motor assays (cylinder, open-field, and apomorphine rotation tests). Rats with confirmed denervation then received once daily i.p. injections for 15 consecutive days of either L-DOPA/benserazide (25/6.25 mg/kg; LID group) or an equal volume of sterile distilled water (Parkinsonism group). AIMs were video scored at selected intervals throughout the treatment course (days 1, 4, 7, 11, and 13) in the LID group. The sham group was administered sterile distilled water, while naïve animals were tested behaviorally at week 3 without pharmacological intervention; all groups were terminally perfused under high-dose anesthesia at the end of the experiment.

The second design targeted PNN degradation (Figure 1B). Following 6-OHDA lesions and behavioral verification of denervation, rats underwent a second stereotaxic procedure in which ChABC (50 U/mL) or vehicle was delivered into the DLS or M1, yielding four cohorts: DLS–ChABC (n = 8), DLS–vehicle (n = 8), M1–ChABC (n = 8), and M1–vehicle (n = 8). L-DOPA/benserazide (25/6.25 mg/kg/day) treatment was initiated on postoperative day 3. Given that ChABC enzymatic activity persists for ~10 days [84], dyskinesias were assessed on alternate days. On day 13, treatments were concluded, and animals were terminally perfused under deep anesthesia.

### 4.4. Stereotaxic Surgery

Rats were anesthetized with ketamine (80 mg/kg, i.p.; Pfizer Inc., New York, NY, USA) and xylazine (10 mg/kg, i.p.; Bioveta a.s., Ivanovice na Hané, Czech Republic), placed in a stereotaxic frame (Stoelting Co., Wood Dale, IL, USA), and maintained under aseptic conditions. Corneas were protected with artificial tears, and the scalp was shaved, disinfected, and infiltrated locally with prilocaine (1–2 mg/kg, i.d.; Vem İlaç, İstanbul, Türkiye). A midline incision was made, bregma and lambda were identified, and the skull was leveled.

Desipramine hydrochloride (25 mg/kg, i.p.; Sigma-Aldrich, St. Louis, MO, USA) was administered 30 min prior to lesioning to protect noradrenergic projections. For MFB lesions, 6-OHDA hydrobromide (12.5 µg in 5 µL of 0.02% ascorbic acid; Cayman Chemical, Ann Arbor, MI, USA) was infused at anteroposterior (AP): −2.2 mm, mediolateral (ML): −1.4 mm, and dorsoventral (DV): −8.1 mm relative to bregma [88]. After the infusion, the needle was left in place for 3 min and then withdrawn gradually. The scalp was sutured, and postoperative analgesia was provided with flunixin (2.5 mg/kg, s.c.; Vilsan İlaç, Ankara, Türkiye).

ChABC (50 U/mL; Sigma-Aldrich, St. Louis, MO, USA) or vehicle (3 µL/per site) was stereotaxically injected into the DLS (AP +1.0, ML −3.2, DV −4.7 mm) and M1 (AP +1.8, ML −2.5, DV −1.3 mm).

### 4.5. Behavioral Tests

All behavioral assessments were performed between 12:00 and 16:00. Animals were acclimated to the testing room for 2 h prior to testing. Scoring was conducted by a single experimenter blinded to group allocation.

#### 4.5.1. Cylinder Test

Motor asymmetry was quantified using the cylinder test. Rats were placed—without prior habituation—into a glass cylinder (21 cm diameter × 34 cm height) housed within an opaque black enclosure, and spontaneous exploration was video-recorded for 6 min. Forelimb use was quantified as the proportion of independent wall contacts made by each paw during rearing and expressed as a percentage of total contacts. The paw contralateral to the lesion was designated as the affected limb. Animals with affected forelimb use below 20% were considered denervated.

#### 4.5.2. Open-Field Motor Activity

Spontaneous locomotion was quantified in transparent Plexiglas arenas (40 × 40 × 40 cm^3^) instrumented with infrared beam arrays (Commat Ltd., Ankara, Türkiye). Animals were recorded for 10 min. Outcome measures were stereotypic, ambulatory, and resting (% of session time), distance (cm; total path length), and horizontal activity counts (total horizontal beam breaks). Arenas were cleaned with 70% ethanol between trials.

#### 4.5.3. Apomorphine-Induced Rotation

To verify nigrostriatal denervation, an apomorphine challenge was performed on post-lesion day 21 as a qualitative cross-check alongside the cylinder test. Rats received apomorphine hydrochloride (0.5 mg/kg, i.p.; Sigma-Aldrich, St. Louis, MO, USA), and rotational behavior was video-recorded. After a 10 min post-injection latency, behavior was monitored for 30 min. The presence of sustained contralateral circling episodes was used qualitatively to confirm denervation; animals exhibiting at least six rotations per min were classified as having more than 90% denervation, consistent with prior work [89].

#### 4.5.4. Abnormal Involuntary Movements (AIMs)

Dyskinesia severity and phenotype were assessed using AIMs scoring. Following L-DOPA/benserazide administration, animals were recorded for 120 min, and AIMs were scored in 1 min epochs at 20 min intervals (20–120 min post-injection). Assessments were performed on treatment days 1, 4, 7, 11, and 13; in ChABC cohorts, AIMs were obtained on days 1, 3, 5, 7, 9, 11, and 13. Subcomponents of AIMs were locomotor (contralateral rotations), axial (contralateral head and neck torsion), limb (contralateral forelimb jerks and hindlimb dystonia), and orolingual stereotypies (vacuum chewing and contralateral tongue protrusions). Each subcomponent was scored on a scale from 0 to 4 (0 absent, 4 continuous and resistant to interruption). Total AIMs scores equaled the sum of subcomponent scores per time point. Outcomes were reported as locomotor versus the ALO composite (axial, limb, and orolingual), and each ALO component was also presented individually [90].

### 4.6. Tissue Preparation

Following completion of treatments, rats were perfused under terminal anesthesia with 4% paraformaldehyde (PFA). Brains were post-fixed in 4% PFA for 24 h, then cryoprotected in 15% and 30% sucrose in PBS at 4 °C until sinking. Tissues were subsequently embedded in O.C.T. compound (Tissue Plus, Scigen, Gardena, CA, USA), snap-frozen in liquid nitrogen, and stored at −80 °C until sectioning. Coronal free-floating 40 µm sections were cut on a sliding microtome (Leica Microsystems, Wetzlar, Germany) and stored in a cryoprotectant solution (ethylene glycol/glycerol/PBS) at −20 °C.

### 4.7. Immunofluorescence Staining

Free-floating sections were processed for dual immunofluorescence staining of PNNs and PV-INs. Sections were rinsed 3 × 5 min in Tris-buffered saline (TBS), subjected to heat-induced epitope retrieval in sodium citrate (85 °C, 15 min), and equilibrated for 30 min at room temperature (RT). Blocking was performed for 3 h in TBS containing 0.3% Triton X-100 (TBS-T), 5% BSA (Merck Millipore, Burlington, MA, USA), and 1% normal goat serum (NGS; Jackson ImmunoResearch, West Grove, PA, USA). Sections were then incubated overnight at 4 °C with biotinylated WFA (1:500; Sigma-Aldrich, St. Louis, MO, USA, Cat. L1516) prepared in TBS-T supplemented with 2% BSA and 1% NGS. After washing, sections were incubated for 2 h at RT with Streptavidin–Alexa Fluor 488 (1:500; Thermo Fisher/Invitrogen, Waltham, MA, USA, Cat. S11223), followed by 5 × 5 min washes in TBS and re-blocking for 1 h. They were subsequently incubated overnight at 4 °C with a rabbit anti-PV antibody (1:500; Proteintech, Rosemont, IL, USA, Cat. 29312-1-AP), and after rinses, incubated for 2 h at RT with a Cy3-conjugated goat anti-rabbit IgG (1:500; Jackson ImmunoResearch, West Grove, PA, USA, Cat. 111-165-144). Finally, sections were washed, mounted on poly-L-lysine–coated slides, and coverslipped with a mounting medium consisting of PBS, glycerol, and Hoechst 33258 (1:1000; Thermo Fisher/Invitrogen, Waltham, MA, USA, Cat. H3569), a nuclear stain.

### 4.8. Imaging and Image Analysis

Imaging was conducted in a subset of the behavioral cohort (n = 6 animals per group). For each animal, three coronal sections were selected at anterior, middle, and posterior striatal levels according to the Paxinos & Watson atlas (+2.28 to +0.12 mm from bregma) (Figure 1C) [88]. Regions of interest (ROIs) were sampled in the DLS, DMS, VS, and motor cortices M1/M2: two ROIs in DLS, one in DMS, one in VS, and two ROIs each in M1 and M2. Due to occasional tissue loss or image quality issues, the final datasets comprised 4–6 ROIs per animal for DLS and M1/M2 cortices, and 2–3 ROIs per animal for DMS and VS.

PV-INs and PNNs were imaged using a Leica TCS SP8 confocal microscope (Leica Microsystems, Wetzlar, Germany) equipped with a 25× water-immersion objective. Images were acquired as single optical sections using 405, 488, and 552 nm excitation. The optical section thickness was 7.20 μm (pinhole 1.0 AU). All acquisition parameters (laser power, detector gain, offset, pinhole size, scanning speed, and pixel resolution) were kept strictly identical across animals and groups to ensure reliable cross-group comparisons. ROIs (250 × 250 μm; 512 × 512 pixels; 0.489 μm/pixel) were captured as predefined sampling windows during acquisition. This single-plane approach is widely used for density-based PNN and PV-IN quantification and aligns with established protocols employing single-section confocal imaging [91].

All analyses were performed on raw confocal images, without any additional preprocessing steps.. Cell counting was performed manually in Fiji/ImageJ (version 1.54; National Institutes of Health, Bethesda, MD, USA) using the cell counter tool by an investigator blinded to group identity. Counted objects were defined as clearly demarcated WFA^+^ or PV^+^ somata within the acquisition-defined ROI. Subpopulations (WFA^+^/PV^−^, WFA^−^/PV^+^, WFA^+^/PV^+^) were classified based on spatial overlap within the same optical section [92]. Fluorescence intensity values for WFA and PV were obtained from the same ROIs using the licensed Polygon AI by Rewire (version 3.1; Rewire Inc., Portland, OR, USA) platform, which computes background-subtracted mean fluorescence intensity (mean–background) for each channel.

### 4.9. Statistical Analysis

All statistical analyses were performed using GraphPad Prism v9 (GraphPad Software, San Diego, CA, USA). Data distribution for each dataset was assessed using the Shapiro–Wilk test. Data are reported as mean ± SEM. For two-group comparisons, unpaired Student’s *t*-tests or Mann–Whitney U tests were used depending on normality. For comparisons involving three or more independent groups, one-way ANOVA with Tukey’s post hoc test or Kruskal–Wallis tests with Dunn’s multiple comparisons were applied. Microscopy-derived metrics (cell densities, subpopulation distributions, and fluorescence intensities) were analyzed at the ROI level as plotted in the Figure 2, Figure 3, Figure 6 and Figure 7.

Longitudinal analyses were restricted to the behavioral AIMs dataset. Day-to-day changes in dyskinesias were analyzed using one-way repeated-measures ANOVA, and group × day interactions with two-way ANOVA. A mixed-effects model (REML) with Šídák correction was used only for partially unbalanced longitudinal AIMs data (DLS-ChABC vs. DLS-vehicle). The specific statistical test used for each Figure is indicated in the corresponding figure legend.

## Figures and Tables

**Figure 1 ijms-26-11726-f001:**
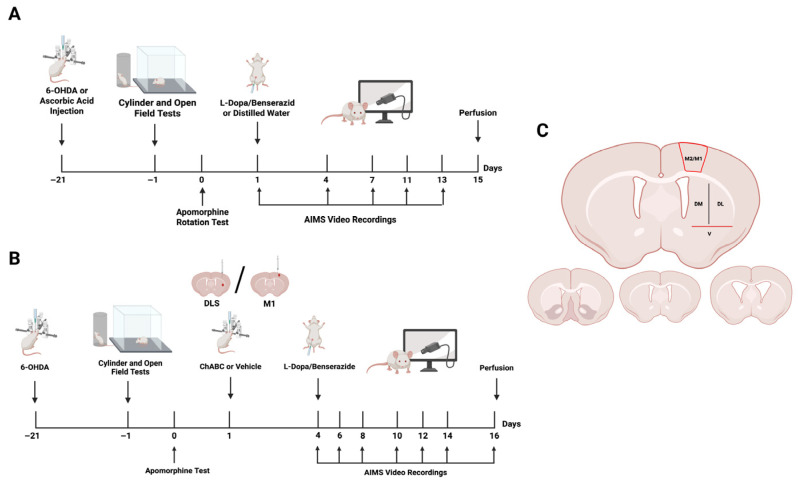
Experimental design and regions of interest. (**A**) Schematic timeline of the LID model. Behavioral assessments (cylinder and open-field tests) were followed by daily L-DOPA/benserazide or distilled water administration, AIMs video recordings, and perfusion. (**B**) Experimental timeline for ChABC or vehicle application. Following the lesion and baseline behavioral testing, ChABC or vehicle was microinjected into the DLS or M1, followed by L-DOPA/benserazide treatment, AIMs recordings, and perfusion. (**C**) Schematic representation of the anatomical regions analyzed for confocal imaging (top) and corresponding coronal levels of the striatum (bottom). Abbreviations: AIMs, abnormal involuntary movements; ChABC, chondroitinase ABC; DL, dorsolateral; DLS, dorsolateral striatum; DM, dorsomedial; LID, L-DOPA-induced dyskinesia; M1, M1 motor cortex; V, ventral; 6-OHDA, 6-hydroxydopamine (Created with BioRender.com).

**Figure 2 ijms-26-11726-f002:**
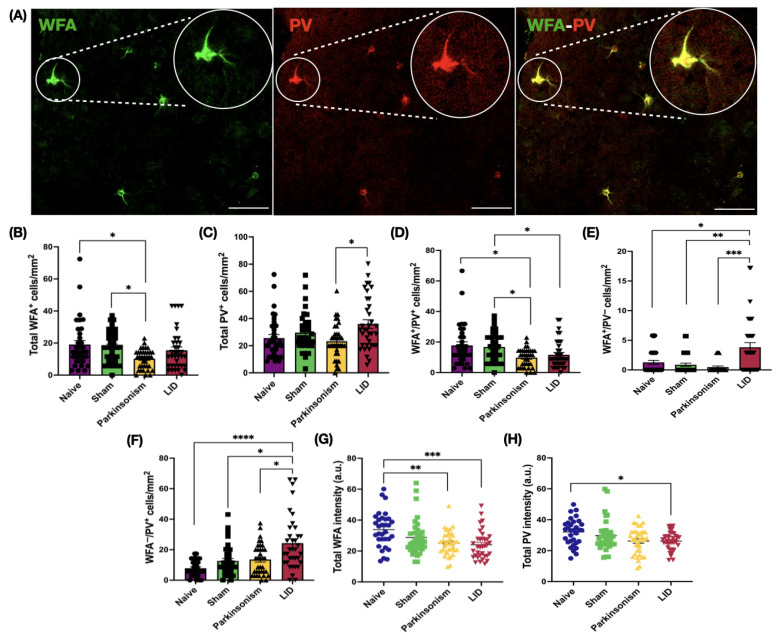
PNN and PV-INs quantification in DLS. (**A**) Representative confocal images from the DLS showing WFA (green, left), PV (red, middle), and merged (right) channels. (**B**) Total WFA cell density (H(3) = 11.85, *p* = 0.0079; naïve vs. Parkinsonism, *p* = 0.0424; sham vs. Parkinsonism, *p* = 0.0102). (**C**) Total PV cell density (H(3) = 10.45, *p* = 0.0151; Parkinsonism vs. LID, *p* = 0.035). (**D**) WFA^+^/PV^+^ cell density (H(3) = 15.08, *p* = 0.0017; naïve vs. Parkinsonism, *p* = 0.0415; sham vs. Parkinsonism, *p* = 0.0109; sham vs. LID, *p* = 0.0385). (**E**) WFA^+^/PV^−^ cell density (H(3) = 20.76, *p* = 0.0001; naïve vs. LID, *p* = 0.0418; sham vs. LID, *p* = 0.0027; Parkinsonism vs. LID *p* = 0.0001). (**F**) WFA^−^/PV^+^ cell density (H(3) = 24.69, *p* < 0.0001; naïve vs. LID, *p* < 0.0001; sham vs. LID, *p* = 0.0136; Parkinsonism vs. LID, *p* = 0.0326). (**G**) Total WFA intensity (H(3) = 17.19, *p* = 0.0006; naïve vs. Parkinsonism, *p* = 0.0076; naïve vs. LID, *p* = 0.0006). (**H**) Total PV intensity (H(3) = 9.86, *p* = 0.0198; naïve vs. LID, *p* = 0.0245). Data are presented as mean ± SEM (each dot represents one ROI). n = 6 animals per group. Detailed data can be found in Appendix A. Kruskal–Wallis test followed by Dunn’s multiple comparisons; * *p* < 0.05, ** *p* < 0.01, *** *p* < 0.001, and **** *p* < 0.0001. Scale bar = 100 µm. Abbreviations: PNN, perineuronal net; PV-IN, parvalbumin interneuron; WFA, Wisteria floribunda agglutinin; PV, parvalbumin; DLS, dorsolateral striatum; LID, L-DOPA-induced dyskinesia; ROI, region of interest.

**Figure 3 ijms-26-11726-f003:**
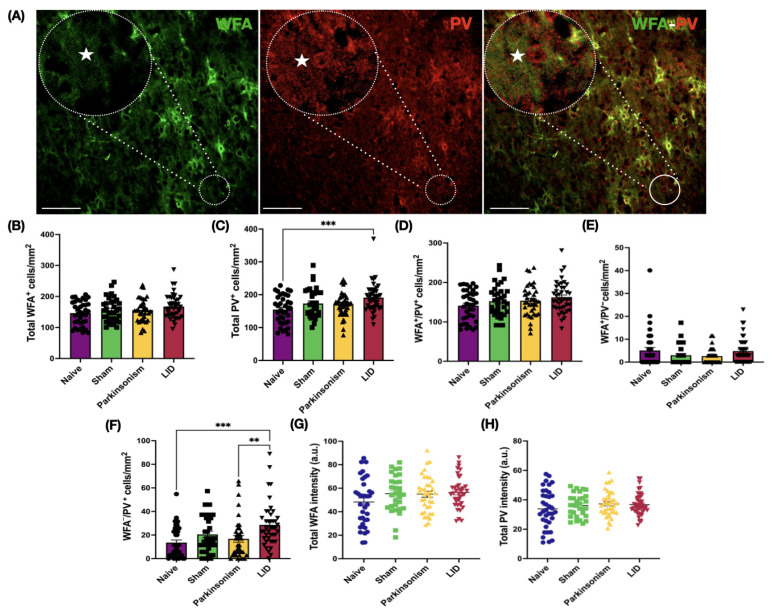
PNN and PV-INs quantification in M1 motor cortex. (**A**) Representative confocal images from the M1 showing WFA (green, left), PV (red, middle), and merged (right) channels. The white star marks a PV-IN lacking WFA enwrapment. (**B**) Total WFA cell density. (**C**) Total PV cell density (H(3) = 11.82, *p* = 0.008; naïve vs. LID, *p* = 0.004). (**D**) WFA^+^/PV^+^ cell density. (**E**) WFA^+^/PV^−^ cell density. (**F**) WFA^−^/PV^+^ cell density (H(3) = 19, *p* = 0.0003; naïve vs. LID *p* = 0.0004; Parkinsonism vs. LID, *p* = 0.0043). (**G**) Total WFA intensity. (**H**) Total PV intensity. Data are presented as mean ± SEM (each dot represents one ROI). n = 6 animals per group. Detailed data can be found in Appendix A. Kruskal–Wallis test followed by Dunn’s multiple comparisons; (**G**,**H**) one-way ANOVA; ** *p* < 0.01, *** *p* < 0.001. Scale bar = 100 µm. Abbreviations: PNN, perineuronal net; PV-IN, parvalbumin interneuron; WFA, Wisteria floribunda agglutinin; LID, L-DOPA-induced dyskinesia; ROI, region of interest.

**Figure 4 ijms-26-11726-f004:**
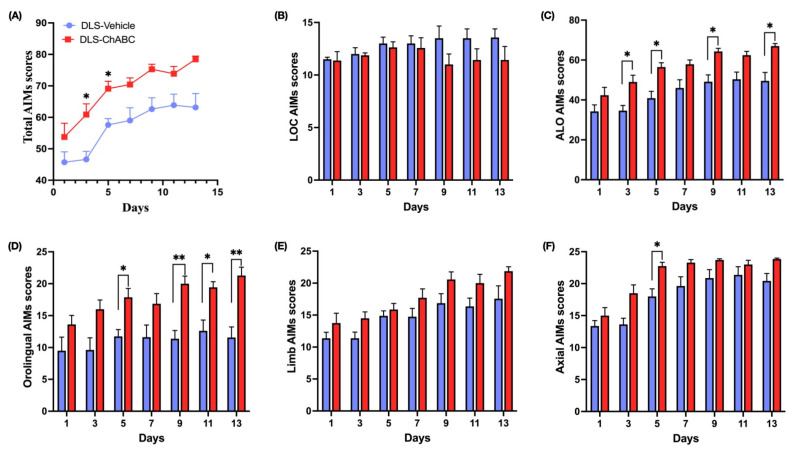
AIMs and subcomponent analyses following DLS-ChABC administration. (**A**) Total AIMs scores (F(1,14) = 14.67, *p* = 0.0018; DLS-ChABC vs. DLS-vehicle on day 3, *p* = 0.037; day 5, *p* = 0.015). (**B**) Locomotor AIMs scores. (**C**) ALO AIMs scores (F(1,14) = 18.43, *p* = 0.0007; DLS-ChABC vs. DLS-vehicle on day 3, *p* = 0.03; day 5, *p* = 0.017; day 9, *p* = 0.017; day 13, *p* = 0.037). (**D**) Orolingual AIMs scores (F(1,14) = 19.02, *p* = 0.0007; DLS-ChABC vs. DLS-vehicle on day 5, *p* = 0.026; day 9, *p* = 0.002; day 11, *p* = 0.033; day 13, *p* = 0.0049). (**E**) Limb AIMs scores. (**F**) Axial AIMs scores (F(1,14) = 8.98, *p* = 0.0096; DLS-ChABC vs. DLS-vehicle on day 5, *p* = 0.032). Data are presented as mean ± SEM (n = 8 animals per group). A mixed-effects model (time × treatment, REML) with Šídák-corrected post hoc tests; * *p* < 0.05, ** *p* < 0.01. Abbreviations: AIMs, abnormal involuntary movements; ALO, axial-limb-orolingual; LOC, locomotor.

**Figure 5 ijms-26-11726-f005:**
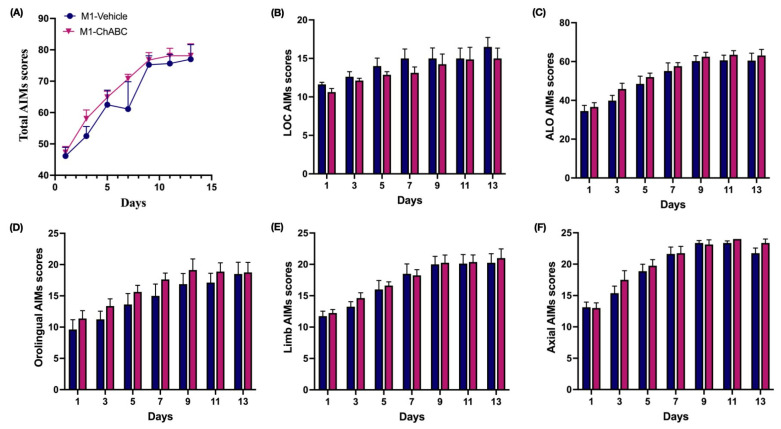
AIMs and subcomponent analyses following M1-ChABC administration. (**A**) Total AIMs scores. (**B**) Locomotor AIMs scores. (**C**) ALO AIMs scores. (**D**) Orolingual AIMs scores. (**E**) Limb AIMs scores. (**F**) Axial AIMs scores. Data are represented as mean ± SEM (n = 8 per group). Two-way repeated-measures ANOVA (time × treatment) with Šídák’s multiple comparisons test. Abbreviations: AIMs, abnormal involuntary movements; ALO, axial-limb-orolingual; LOC, locomotor.

**Figure 6 ijms-26-11726-f006:**
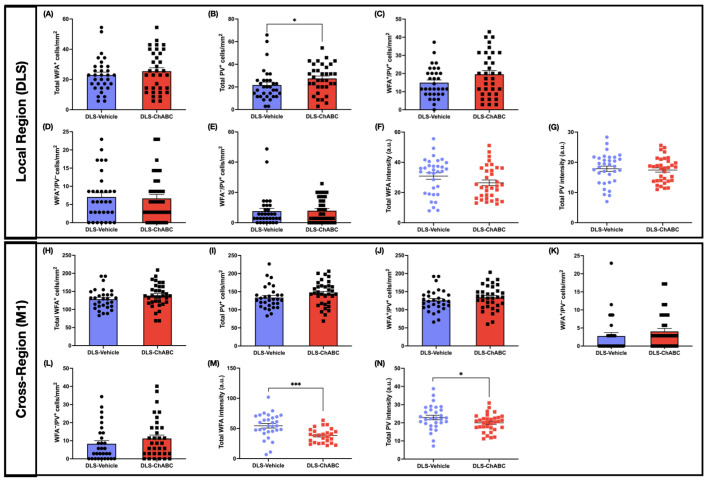
PNN and PV-IN quantification following DLS–ChABC administration in the injection-site region (DLS) and the corresponding cortical region (M1). (**A**–**E**) represent PNN–PV-IN cell densities in the DLS; (**F**,**G**) show PNN–PV-IN intensities in the DLS. (**H**–**L**) depict PNN–PV-IN cell densities in the M1; (**M**,**N**) show PNN–PV-IN intensities in the M1. (**A**) Total WFA cell density. (**B**) Total PV cell density (*p* = 0.023). (**C**) WFA^+^/PV^+^ cell density. (**D**) WFA^+^/PV^−^ cell density. (**E**) WFA^−^/PV^+^ cell density. (**F**) Total WFA intensity. (**G**) Total PV intensity. (**H**) Total WFA cell density. (**I**) Total PV cell density. (**J**) WFA^+^/PV^+^ cell density. (**K**) WFA^+^/PV^−^ cell density. (**L**) WFA^−^/PV^+^ cell density. (**M**) Total WFA intensity (*p* = 0.0007). (**N**) Total PV intensity (*p* = 0.0498). Data are presented as mean ± SEM (each dot represents one ROI). n = 6 animals per group. Detailed data can be found in Appendix A. (**A**,**B**,**D**–**F**,**I**,**K**,**L**) Mann–Whitney U test; (**C**,**G**,**H**,**J**,**M**,**N**) unpaired *t*-test; * *p* < 0.05, *** *p* < 0.001. Abbreviations: PNN, perineuronal net; PV-IN, parvalbumin interneuron; WFA, Wisteria floribunda agglutinin; PV, parvalbumin; DLS, dorsolateral striatum; M1, M1 motor cortex; ChABC, Chondroitinase ABC; ROI, region of interest.

**Figure 7 ijms-26-11726-f007:**
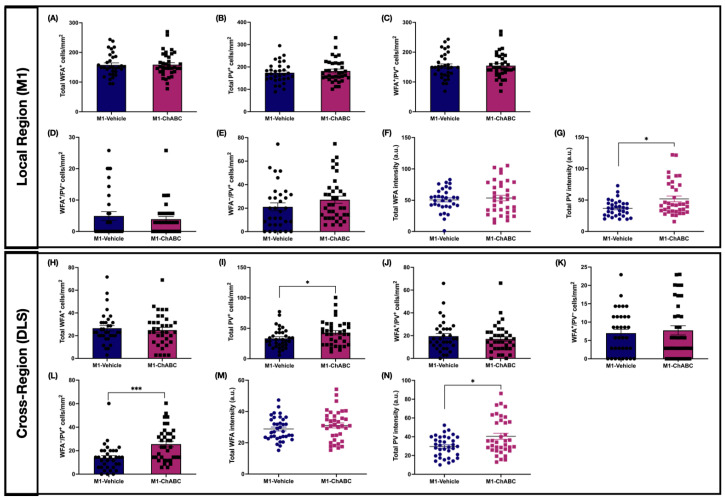
PNN and PV-IN quantification following M1–ChABC administration in the injection-site region (M1) and the corresponding striatal region (DLS). (**A**–**E**) represent PNN–PV-IN cell densities in the M1; (**F**,**G**) show PNN–PV-IN intensities in the M1. (**H**–**L**) depict PNN–PV-IN cell densities in the DLS; (**M**,**N**) show PNN–PV-IN intensities in the DLS. (**A**) Total WFA cell density. (**B**) Total PV cell density. (**C**) WFA^+^/PV^+^ cell density. (**D**) WFA^+^/PV^−^ cell density. (**E**) WFA^−^/PV^+^ cell density. (**F**) Total WFA intensity. (**G**) Total PV intensity (*p* = 0.0251). (**H**) Total WFA cell density. (**I**) Total PV cell density (*p* = 0.0403). (**J**) WFA^+^/PV^+^ cell density. (**K**) WFA^+^/PV^−^ cell density. (**L**) WFA^−^/PV^+^ cell density (*p* = 0.0001). (**M**) Total WFA intensity. (**N**) Total PV intensity (*p* = 0.0458). Data are presented as mean ± SEM (each dot represents one ROI). n = 6 animals per group. Detailed data can be found in Appendix A. (**D**,**E**,**G**,**H**,**J**–**L**,**N**) Mann–Whitney U test; (**A**–**C**,**F**,**I**,**M**) unpaired *t*-test; * *p* < 0.05, *** *p* < 0.001. Abbreviations: PNN, perineuronal net; PV-IN, parvalbumin interneuron; WFA, Wisteria floribunda agglutinin; PV, parvalbumin; DLS, dorsolateral striatum; M1, M1 motor cortex; ChABC, Chondroitinase ABC; ROI, region of interest.

## Data Availability

The original contributions presented in this study are included in the article/Appendix A. Further inquiries can be directed to the corresponding author(s).

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
