# Peer review of "Remodeling of Perineuronal Nets in the Striato-Cortical Axis in L-DOPA-Induced Dyskinesia Rat Model"

_ijms, 2025, doi:10.3390/ijms262311726_

Round 1
Reviewer 1 Report
Comments and Suggestions for Authors
- The study is promising and the manuscript is generally well-written. However, several minor points need to be addressed and clarified before the manuscript can be considered for publication.
- Results: Move all exact numerical data (means, SEMs, p-values, ROI counts) to
Tables or Supplementary Material. - Keep Results text conceptual, describing direction/trends and significance.
- Discussion: The Discussion repeats large blocks of the Results. Focus on interpretation, not recapitulation. Strengthen the conceptual model linking DLS–M1 plasticity and PNN fragility.
- Figures: Ensure figure legends are shorter and focus on interpretation.
- Terminology Consistency: Use:
- “PNN–PV associations” consistently
- “PV-INs” (not PV interneurons / PV+ cells interchangeably)
- “parkinsonism” vs “Parkinsonian state” (choose one)
Author Response
Comment 1: The study is promising and the manuscript is generally well-written. However, several minor points need to be addressed and clarified before the manuscript can be considered for publication.
Response 1: We thank the reviewer for the careful assessment of our manuscript. We appreciate the constructive feedback and have carefully addressed all minor points raised. All corresponding revisions are now incorporated into the revised manuscript and are highlighted in the annotated version.
Comment 2: Results: Move all exact numerical data (means, SEMs, p-values, ROI counts) to Tables or Supplementary Material.
Response 2: We thank the reviewer for this helpful suggestion. In accordance with the recommendation, all numerical values related to cellular metrics (means, SEMs, p-values, and ROI counts) have been removed from the Results text and transferred to the Supplementary Tables for clarity and accessibility. For the behavioral datasets, the explicit numerical values were removed from the main text to improve readability, while the data remains fully represented in the corresponding figures. This adjustment allowed the Results section to focus more clearly on the main findings, while the detailed quantitative data are now presented in an organized and accessible format in the Supplementary Material.
Comment 3: Keep Results text conceptual, describing direction/trends and significance.
Response 3 Thank you for this comment. In accordance with your suggestion, we revised the Results section to remove discussion-like statements. Since deleted sentences cannot be highlighted, the sentence immediately preceding each deletion has been highlighted in the revised manuscript to indicate where the changes were made. We kept only one short and neutral summary sentence at the end of each subsection to outline the main pattern of the findings. This preserves the descriptive nature of the Results while maintaining clarity for the reader.
Comment 4: Discussion: The Discussion repeats large blocks of the Results. Focus on interpretation, not recapitulation. Strengthen the conceptual model linking DLS–M1 plasticity and PNN fragility.
Response 4: Thank you for this insightful comment. In line with your suggestion, we removed all discussion-like statements from the Results section. Once these were eliminated, the overlapping portions in the Discussion naturally disappeared. We also refined and clarified the Discussion to maintain focus on interpretation. We now emphasized the conceptual framework of DLS–M1 interactions and PNN fragility.
Comment 5: Figures: Ensure figure legends are shorter and focus on interpretation.
Response 5: Thank you for this comment. We went through all figure legends and simplified them as suggested. The ROI counts were removed from the legends and placed into the Supplementary Tables. We also deleted non-significant p-values to avoid unnecessary clutter. The legends now give only the essential information needed to follow the figures, without repeating detailed numerical results.
Comment 6: Terminology Consistency: Use:“PNN–PV associations” consistently,“PV-INs” (not PV interneurons / PV+ cells interchangeably),“parkinsonism” vs “Parkinsonian state” (choose one)
Response 6: Thank you for pointing this out. We revised the manuscript to ensure consistent terminology throughout. “PNN–PV associations,” “PV-INs,” and “parkinsonism” are now used uniformly across the text where applicable.
Reviewer 2 Report
Comments and Suggestions for Authors
The manuscript titled “Remodeling of Perineuronal Nets in the Striato-Cortical Axis in L-DOPA-Induced Dyskinesia Rat Model” by Nedime Tugce Bilbay and collabirators is very well written, organized, and scientifically solid. The authors explore how alterations in perineuronal nets (PNNs) across the striato-cortical circuitry may contribute to the development of L-DOPA–induced dyskinesia (LID). Using a unilateral 6-OHDA rat model followed by chronic L-DOPA administration, they carry out a thorough analysis of PNN–parvalbumin interneuron (PV-IN) interactions in different striatal regions and in motor cortical areas.
Their findings show that dopamine depletion leads to a marked reduction in both the density and intensity of PNNs in the dorsolateral striatum (DLS). Although L-DOPA partly restores PNN density, the overall integrity of these structures remains compromised. The authors also report a notable shift from the canonical WFA⁺/PV⁺ phenotype toward an expanded WFA⁻/PV⁺ population in both the DLS and M1 cortex, pointing to a more plastic and potentially unstable microcircuit environment associated with LID.
To probe causality, the study uses region-specific chondroitinase ABC (ChABC) digestion. Degrading PNNs in the DLS worsens dyskinesia—especially the orolingual component—while ChABC applied in M1 produces measurable cellular effects without aggravating behavior. Overall, the results highlight the DLS as a key site where PNN vulnerability can intensify dyskinetic symptoms and reveal coordinated PNN–PV-IN remodeling along the DLS–M1 axis. The work suggests that preserving extracellular matrix integrity could be a promising strategy to counteract maladaptive plasticity in LID.
Author Response
Comments: The manuscript titled “Remodeling of Perineuronal Nets in the Striato-Cortical Axis in L-DOPA-Induced Dyskinesia Rat Model” by Nedime Tugce Bilbay and collabirators is very well written, organized, and scientifically solid. The authors explore how alterations in perineuronal nets (PNNs) across the striato-cortical circuitry may contribute to the development of L-DOPA–induced dyskinesia (LID). Using a unilateral 6-OHDA rat model followed by chronic L-DOPA administration, they carry out a thorough analysis of PNN–parvalbumin interneuron (PV-IN) interactions in different striatal regions and in motor cortical areas.
Their findings show that dopamine depletion leads to a marked reduction in both the density and intensity of PNNs in the dorsolateral striatum (DLS). Although L-DOPA partly restores PNN density, the overall integrity of these structures remains compromised. The authors also report a notable shift from the canonical WFA⁺/PV⁺ phenotype toward an expanded WFA⁻/PV⁺ population in both the DLS and M1 cortex, pointing to a more plastic and potentially unstable microcircuit environment associated with LID.
To probe causality, the study uses region-specific chondroitinase ABC (ChABC) digestion. Degrading PNNs in the DLS worsens dyskinesia—especially the orolingual component—while ChABC applied in M1 produces measurable cellular effects without aggravating behavior. Overall, the results highlight the DLS as a key site where PNN vulnerability can intensify dyskinetic symptoms and reveal coordinated PNN–PV-IN remodeling along the DLS–M1 axis. The work suggests that preserving extracellular matrix integrity could be a promising strategy to counteract maladaptive plasticity in LID.
Response: We sincerely thank the reviewer for the thoughtful evaluation and encouraging comments regarding our work. We appreciate the recognition of our study’s scientific rigor and the relevance of our findings. No further revisions were requested.
Reviewer 3 Report
Comments and Suggestions for Authors
The submitted article addresses a highly relevant and under-explored issue: the role of perineuronal nets in plasticity within the context of neurodegeneration and neurocompensation in varios brain regions. The study appears to be well-designed and methodologically sound.
However, I have several questions for the authors that require clarification to fully evaluate and interpretation of the data:
- Microscopy methodology: could you describe in more detail the protocol for acquiring and processing brain slice images? Specifically, please provide the scanning depth, the optical section thickness, the specific FIJI plugins used, and the key image analysis parameters.
- Statistical analysis: the 'Statistical Analysis' section states that a mixed-effects model (REML) with Šidák's correction was used. Could you please clarify whether this model was applied for the analysis of the microscopy data presented in Figures 3, 6, etc.? Given that individual ROI values are plotted, it is crucial to know how the statistical analysis was performed: mean values for each animal in the group or a mixed-effects model. Please specify this for each figure.
Author Response
The submitted article addresses a highly relevant and under-explored issue: the role of perineuronal nets in plasticity within the context of neurodegeneration and neurocompensation in varios brain regions. The study appears to be well-designed and methodologically sound.
However, I have several questions for the authors that require clarification to fully evaluate and interpretation of the data:
Comment 1: Microscopy methodology: could you describe in more detail the protocol for acquiring and processing brain slice images? Specifically, please provide the scanning depth, the optical section thickness, the specific FIJI plugins used, and the key image analysis parameters.
Response 1: We thank the reviewer for this valuable comment. To improve clarity and reproducibility, we expanded and highlighted the Microscopy and Image Analysis section. The revised text now provides additional details on the confocal acquisition parameters, the use of single-plane imaging, the criteria used to identify WFA⁺ and PV⁺ somata, and our FIJI-based manual counting workflow. We also clarified that all analyses were performed on raw images without preprocessing. Additionally, we incorporated two methodological references that closely match our imaging approach.
Comment 2: Statistical analysis: the 'Statistical Analysis' section states that a mixed-effects model (REML) with Šidák's correction was used. Could you please clarify whether this model was applied for the analysis of the microscopy data presented in Figures 3, 6, etc.? Given that individual ROI values are plotted, it is crucial to know how the statistical analysis was performed: mean values for each animal in the group or a mixed-effects model. Please specify this for each figure.
Response 2: We thank the reviewer for raising this important point. We expanded and highlighted the Statistical Analysis section. Briefly, mixed-effects modeling (REML with Šidák’s correction) was used only for the longitudinal AIMs data (DLS-ChABC vs. DLS-vehicle). All microscopy-based quantifications were analyzed at the ROI level, as plotted in the figures. Depending on data distribution and group number, we used one-way ANOVA or Kruskal–Wallis tests for comparisons involving more than two groups, and unpaired t-tests or Mann–Whitney tests for pairwise comparisons.